# Tumour-informed liquid biopsies to monitor advanced melanoma patients under immune checkpoint inhibition

Christopher Schroeder [1,2], Sergios Gatidis [3], Olga Kelemen[1], Leon Schütz [1], Irina Bonzheim [4], Francesc Muyas[1], Peter Martus[5], Jakob Admard [1,6], Sorin Armeanu-Ebinger[1], Brigitte Gückel[3], Thomas Küstner [3], Claus Garbe[7], Lukas Flatz[7], Christina Pfannenberg[3], Stephan Ossowski [1,2,6,8] & Andrea Forschner [7] ✉

Immune checkpoint inhibitors (ICI) have significantly improved overall survival in melanoma patients. However, 60% experience severe adverse events and early response markers are lacking. Circulating tumour DNA (ctDNA) is a promising biomarker for treatment-response and recurrence detection. The prospective PET/LIT study included 104 patients with palliative combined or adjuvant ICI. Tumour-informed sequencing panels to monitor 30 patient-specific variants were designed and 321 liquid biopsies of 87 patients sequenced. Mean sequencing depth after deduplication using UMIs was 6000x and the error rate of UMI-corrected reads was $2.47 \times 10^{-4}$. Variant allele fractions correlated with PET/CT MTV (rho=0.69), S100 (rho=0.72), and LDH (rho=0.54). A decrease of allele fractions between T1 and T2 was associated with improved PFS and OS in the palliative cohort (p = 0.008 and p < 0.001). ctDNA was detected in 76.9% of adjuvant patients with relapse (n = 10/13), while all patients without progression (n = 9) remained ctDNA negative. Tumour-informed liquid biopsies are a reliable tool for monitoring treatment response and early relapse in melanoma patients with ICI.

Immune checkpoint inhibitors (ICI) have significantly improved overall survival in patients with metastatic melanoma and have recently been approved in the adjuvant setting for patients with stage IIB/C-IV[1–4]. The 5-year overall survival rate in the palliative setting has increased to more than 50% for patients receiving first-line combined ICI with ipilimumab and nivolumab[5]. However, about 60% of the patients treated with combined ICI experience serious immune-related adverse events that can be life-threatening and more than one third of patients has to discontinue treatment due to toxicity[1].

Therefore, early response assessment in patients with combined ICI is urgently needed. Previous studies have shown that a high TMB, presence of *BRAF* mutation, normal serum lactate dehydrogenase, low neutrophil/lymphocyte ratio, Eastern Cooperative Oncology Group performance status 0, and absence of liver metastases are associated with improved treatment response to ICI[5–8].

Imaging techniques such as PET/CT have high sensitivity and specificity for assessing response to ICI or detecting recurrence[9,10]. This includes differentiating between progressive disease and

[1]Institute of Medical Genetics and Applied Genomics, University of Tübingen, Tübingen, Germany. [2]German Cancer Consortium (DKTK), partner site Tübingen, German Cancer Research Center (DKFZ), Heidelberg, Germany. [3]Department of Radiology, Diagnostic and Interventional Radiology, University Hospital Tübingen, Tübingen, Germany. [4]Institute of Pathology and Neuropathology, University Hospital Tübingen, Tübingen, Germany. [5]Institute for Clinical Epidemiology and Applied Biostatistics (IKEaB), Tübingen, Germany. [6]NGS Competence Center Tübingen (NCCT), University of Tübingen, Tübingen, Germany. [7]Department of Dermatology, University Hospital Tübingen, Tübingen, Germany. [8]Institute for Bioinformatics and Medical Informatics (IBMI), University of Tübingen, Tübingen, Germany. ✉e-mail: andrea.forschner@med.uni-tuebingen.de

inflammatory processes or pseudoprogression[11]. However, PET/CT is expensive and cannot be performed frequently or at every centre[12]. Circulating tumour DNA (ctDNA) in plasma is a promising biomarker that can be detected in most tumour patients, especially those with advanced disease[13,14]. ctDNA assays have the advantage of being only minimally invasive and the measurements are easy to integrate into the clinical routine. We and others have shown that plasma ctDNA levels at baseline correlate with disease stage, tumour burden and treatment response to ICI in melanoma patients[15–17].

Although ctDNA is a promising biomarker, detection of low levels in plasma is still challenging, rendering early detection and disease monitoring difficult. Technological advances, such as unique molecular identifiers and ultra-deep sequencing, have significantly increased the sensitivity of next-generation sequencing (NGS) approaches[17,18]. More recently, the combined detection of selected patient-specific somatic mutations, known as tumour-informed approaches, has further improved the detection limit, allowing monitoring of minimal residual disease and the detection of early relapse[19–26].

This prospective, non-interventional study aims to use tumour-informed liquid biopsies to predict response and disease relapse under treatment with ICI in patients with advanced melanoma. Tumour-normal samples were sequenced as part of routine clinical care. Subsequently, liquid biopsies were taken before, during and after treatment to monitor up to 30 patient-specific somatic tumour mutations in circulating cell-free DNA. Liquid biopsy results were compared with standard blood biomarkers and 18F-FDG-PET/CT.

## Results

### Patient cohort, tumour sequencing and plasma samples

We recruited 104 patients with advanced melanoma who were scheduled to receive ICI. Four of these patients were unable to start with ICI, one patient unfortunately died of his tumour before combined ICI could be started, one patient withdrew her consent for adjuvant ICI due to concerns about potential immune-mediated side effects, another patient changed his decision and opted for adjuvant targeted therapy instead of adjuvant ICI and in the fourth patient it turned out that the lung metastasis was found to be caused by another tumour rather than melanoma. Furthermore, sequencing of tumour and normal tissue to identify tumour-specific variants was not possible in seven patients due to low tumour content or empty FFPE blocks. Baseline PET/CT, liquid biopsies, S100 and LDH were obtained prior to initiation of ICI. In the palliative setting, the second PET/CT was usually performed after 4 cycles of combined ICI, which corresponds to ~12 weeks after treatment initiation (Fig. 1). In the adjuvant setting, the

second PET/CT was usually performed after 6 months of treatment, in accordance with the current German guideline, which indicates 6 monthly intervals for staging during the first three years in tumour-free stage III or IV patients[27].

A total of 376 liquid biopsies of 93 patients before, during and after treatment were included in this study. Fifty-five liquid biopsies had to be removed from further analysis due to haemolysis, low DNA quality, low sequencing depth or non-matching SNP fingerprint. As a consequence, six additional patients without liquid biopsy results were excluded. The final cohort consisted of 87 patients, 65 stage IV patients with combined ICI in the palliative setting and 22 stage III patients with adjuvant therapy following surgery of lymph node metastases. All 65 patients with combined ICI in the palliative setting had unresectable stage IV melanoma, less than half of them (43%) received combined ICI as first-line treatment. and more than 50% of the patients had elevated LDH or S100 values at baseline. The adjuvant cohort consisted of tumour-free stage III patients, of whom 15/22 (68%) were stage IIIC/D at the time of ICI initiation. Tumour sequencing identified hotspot mutations in BRAF (V600) and NRAS (Q61, G12, G13) in 25/87 (29%) and 39/87 (45%) of patients respectively. A hotspot mutation in BRAF and/or NRAS was found in 63/87 patients (72%, one patient with a BRAF and NRAS hotspot mutation). Notably, promoter variants in TERT included in the panel design of 26 patients had a comparable sequencing depth to other variants (Supplementary Fig. 1). Details of the final cohort are provided in Table 1 and Supplementary Fig. 2, which shows hotspot and non-hotspot mutations in a subset of selected melanoma driver genes.

### Sequencing analysis of cell-free DNA

For the remaining 87 patients, sequencing data of tumour and blood were available and up to 30 somatic mutations were selected for subsequent monitoring in plasma (Supplementary Fig. 3). For the 321 plasma samples, an average depth of ~6000× after deduplication (keeping only reads with at least one duplicate) was obtained. The mean error rate for reads with at least one duplicate was $2.47 \times 10^{-4}$ (range $4.58 \times 10^{-5} - 1.24 \times 10^{-3}$) and $9.00 \times 10^{-5}$ (range $1.45 \times 10^{-5} - 6.73 \times 10^{-4}$) for reads with at least two duplicates (Supplementary Table 1). Samples within 7 days of treatment initiation and with at least three variants sufficiently covered were available for 60 patients in the combined ICI group and three or more tumour variants were detected in 87% (52/60) of patients. Fig 2 shows examples of patients with multiple liquid biopsies during treatment. These examples illustrate that tumour-informed liquid biopsies allow close monitoring of the allele fractions (AF) of tumour variants in plasma over multiple time points. Patients who benefit from ICI show a clear and rapid

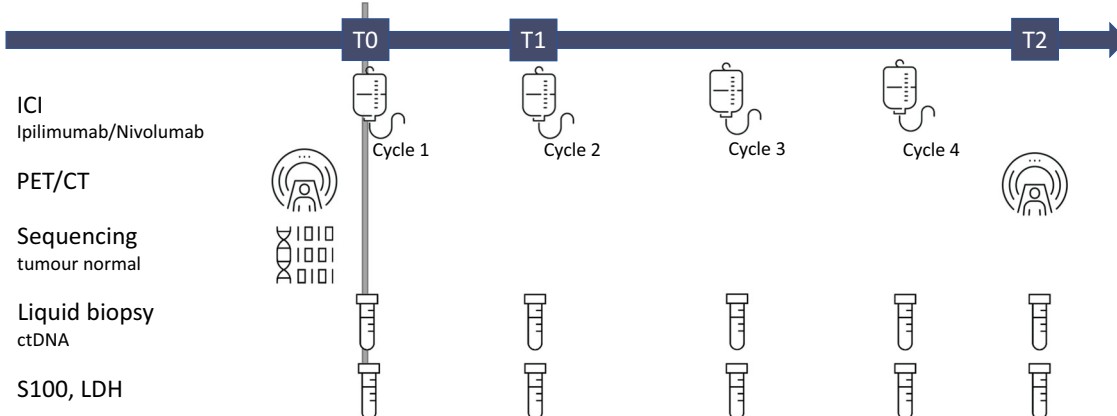

**Fig. 1 | Study design for patients with combined ICI.** PET/CT and sequencing of tumour normal pairs for patients under palliative ICI were done before treatment start. Blood samples were taken at T0, T1 (3 weeks) and T2 (second PET/CT, 12 weeks). Additional blood samples were collected during therapy and follow-up to increase the sensitivity for changes in allele fractions during and after therapy completion.

**Table. 1 | Patient characteristics. Source data are provided as a Source Data file**

| Patient characteristics | | No° patients | % |
|---|---|---|---|
| Total | | 87 | 100 |
| Sex | Female | 38 | 44 |
| | Male | 49 | 56 |
| Age at start of immune checkpoint inhibitor [median; IQR] | | 64; 56–76 years | |
| Melanoma type | | | |
| | Cutaneous | 67 | 77 |
| | Occult | 6 | 7 |
| | Uveal | 1 | 1 |
| | Acral | 7 | 8 |
| | Mucosal | 6 | 7 |
| Adjuvant cohort (Nivolumab/Pembrolizumab) | | 22 | |
| | Stage I + II/III/IV at initial diagnosis | 10/12/0 | |
| | Tumour-free stage III at treatment start | 22 | |
| | Patient alive at last follow-up | 19 | |
| Palliative cohort (Ipilimumab + Nivolumab) | | 65 | |
| | Stage I + II/III/IV at initial diagnosis | 28/22/15 | |
| | Unresectable stage IV at start of combined ICI | 65 | |
| | First line treatment | 28 | |
| | LDH elevated* at start of combined ICI | 34 | |
| | S100 elevated** at start of combined ICI | 38 | |
| | Patient alive at last follow-up | 29 | |

\* >250 U/l, \*\* ≥0.1 µg/l.

decline in ctDNA levels, while progression is associated with increasing ctDNA levels in most cases. Interestingly, a few patients with an exceptionally high tumour uptake on PET/CT and high plasma AFs did not show an increase but stable AFs at progression (Supplementary Fig. 4). This may be related to plasma saturation with ctDNA. In these cases, we frequently observed increasing cell-free DNA concentrations in plasma instead.

## Comparison of ctDNA with established biomarkers

Sequencing results of advanced melanoma patients were correlated with other blood-based biomarkers and imaging data. A total of 188 S100 measurements from 64 patients were available within ±7 days of a liquid biopsy with at least three covered variants. The S100 levels of patients with combined ICI were strongly correlated with the mean variant AFs of tumour variants in cell-free DNA (Spearman's rho 0.72, $t = 8.17$, df = 62, $p = 2.01 \times 10^{-9}$, Fig. 3A). Similarly, LDH levels of 201 measurements in 64 patients with combined ICI within ±7 days of a liquid biopsy correlated moderately with the mean variant AFs (Spearman's rho 0.54, $t = 5.06$, df = 62, $p = 4.12 \times 10^{-4}$, Fig. 3B). Finally, data were available of 90 PET scans of 60 patients with combined ICI within ±21 days of a liquid biopsy. The metabolic tumour volume (MTV) was strongly correlated with the mean variant allele fraction in plasma (Spearman's rho 0.69, $t = 7.26$, df = 58, $p = 1.1 \times 10^{-6}$, Fig. 3C). Similarly, TLG was strongly correlated with the mean variant AF (Spearman's rho 0.67, $t = 6.87$, df = 58, $p = 4.8 \times 10^{-6}$, Supplementary Fig. 5). Liquid biopsies were able to identify more samples with ctDNA compared to S100 or LDH in the cohort of advanced melanoma patients at T0 (Supplementary Fig. 6) and therefore indicated a higher sensitivity compared to the other biomarkers. Patient 27 had two consecutive ctDNA-positive samples while his PET remained negative. This patient's tumour showed no metabolic activity, but disease progression was confirmed by both increasing ctDNA levels and CT-imaging (Fig. 2A). We conclude that this is a rare case of metabolic-negative melanoma metastases on PET.

## Survival analysis

In the group of patients with combined ICI, elevated LDH levels at T0 were significantly correlated with worse PFS ($p = 0.014$, Supplementary Fig. 7B) and OS ($p = 0.030$). Increased tumour mutational burden (TMB) was associated with significant better PFS ($p = 0.014$, Supplementary Fig. 7A). For PET imaging, increasing MTV between T0 and T2 was highly significant for worse OS and PFS ($p = 6.678 \times 10^{-9}$ and $p = 6.973 \times 10^{-6}$). The presence of ctDNA (residual tumour p < 0.05) at T1 was correlated with worse PFS ($p = 0.015$), the presence of ctDNA at T2 was correlated with worse PFS and OS at T2 ($p = 9.056 \times 10^{-4}$ and $p = 0.002$). Decreasing AFs of tumour variants in plasma between T0 and T1 were associated with improved PFS and OS ($p = 0.023$ and $p = 0.029$, Fig. 4A, B), whereas increasing or stable AFs were associated with worse OS and PFS. Furthermore, decreasing AFs between T1 and T2 were highly associated with improved PFS and OS ($p = 0.008$ and $p = 3.712 \times 10^{-4}$, Fig. 4C, D). The hazard ratio for progression with increasing ctDNA allele fractions between T0-T1 in a univariate Cox proportional hazards model was HR 2.46 ($n = 47$, CI 1.11–5.48) for PFS and HR 2.46 ($n = 47$, CI 1.07–5.65) for OS. Similarly, increasing ctDNA allele fractions between T1 and T2 had a hazard ratio for progression of HR 4.45 ($n = 29$, CI 1.35–14.68) for PFS and HR 7.28 ($n = 29$, CI 2.06–25.65) for OS.

In the adjuvant cohort, 13 of 22 patients had progressive disease during follow-up (Fig. 4E). Included were samples with at least three covered variants. Three patients were ctDNA-negative, i.e. less than 3 variants detected in a plasma sample, despite progressive disease: Patient 65 had disease progression in the central nervous system which could have caused the negative result in plasma. Patient 82 had a small cutaneous metastasis of less than 5 mm that was not detected in plasma or PET/CT. Patient 104 had three ctDNA-positive liquid biopsies starting 110 days before progression but turned negative again in follow-up samples. The positive liquid biopsies coincided with the patient's adjuvant radiation therapy which may explain the elevated plasma ctDNA levels. In summary, 10 out of 13 patients (76.9%) had ctDNA-positive results associated with disease progression. When looking at patients who were ctDNA-positive before or within seven days of clinical progression, we excluded three more patients: Patient 68, who did not have a liquid biopsy within the seven days and was ctDNA-negative 39 days before progression (but became positive later). Patient 73, who was ctDNA positive before and after progression, but negative at progression. The sample closest to the progression in this patient had an average depth of less than 1400× (compared to an average depth of 6000× across all liquid biopsies), which may explain the lower sensitivity and the negative result. Patient 75 was negative at progression but positive shortly afterwards at day 8. The positive sample had the highest sequencing depth (>6000×) of his samples and therefore probably the highest sensitivity. In total, we were able to detect ctDNA in plasma of 7 patients before or within 7 days of disease progression. Disease progression was detected up to 133 days before regular imaging (e.g. patient 91). Of the nine patients without progression, none had detectable plasma ctDNA throughout follow-up ($n = 31$ liquid biopsies). Although the results have to be interpreted with caution due to the small cohort size, the presence of ctDNA was associated with a higher relapse rate in our group with adjuvant ICI compared to patients without detectable ctDNA.

## Discussion

The prospective PET/LIT study included 93 patients with advanced melanoma who were receiving palliative combined ICI or adjuvant anti-PD-1 antibodies. Plasma samples were collected before and during treatment (median of 3 samples) and were subsequently analysed in 87 patients. Liquid biopsy monitoring proved to be complementary and, in some cases, superior to established biomarkers such as S100 or LDH and correlated well with the metabolic tumour burden on PET. This allowed for highly sensitive monitoring of treatment response or

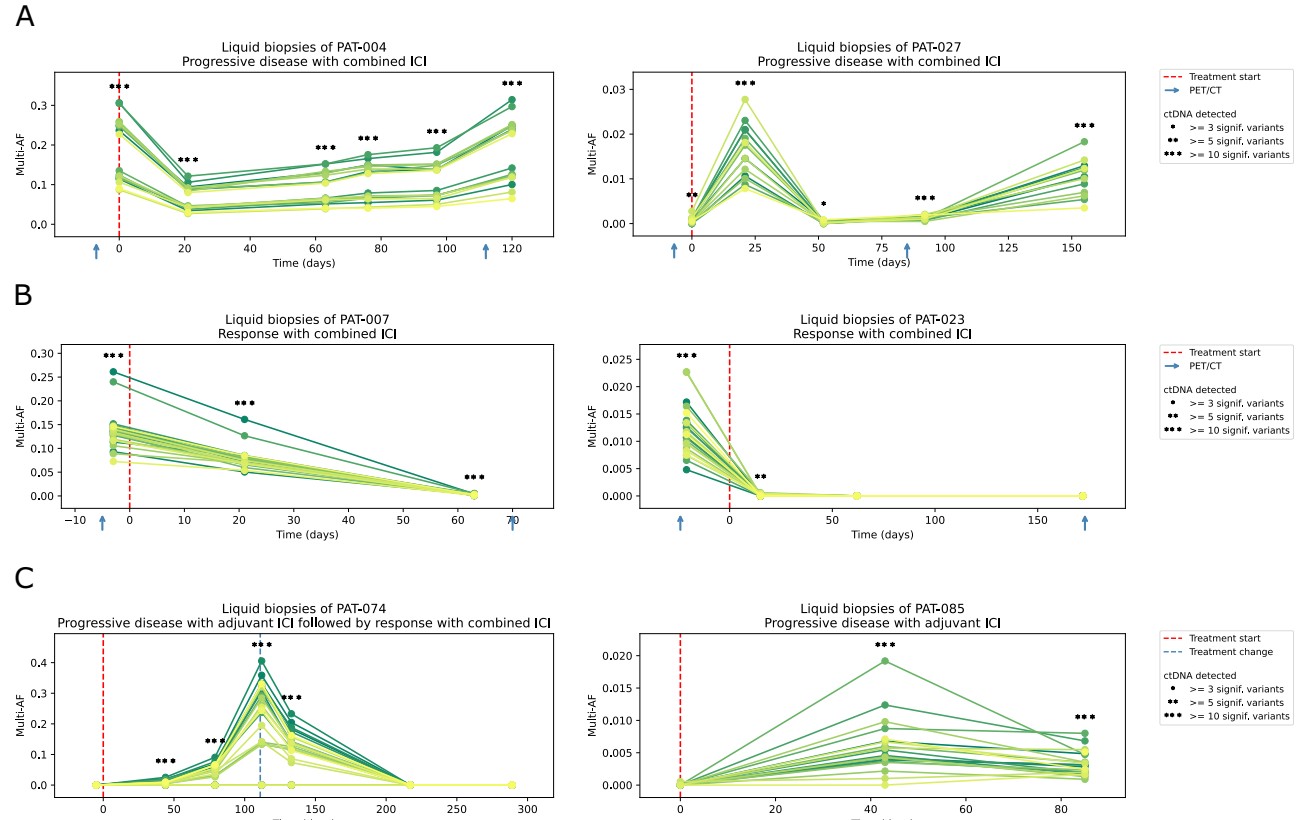

**Fig. 2 | Examples of ctDNA sequencing results of advanced melanoma patients.** **A** shows ctDNA sequencing results (allele fractions of all monitored variants over time) of two patients under combined ICI with progressive disease. An early decrease in variant AF is followed by a subsequent increase. Due to the minimal invasive nature of liquid biopsies, closer monitoring over time can be obtained compared to PET/CT. The different allele fractions of the somatic variants reflect the clonal structure of the primary tumour which remained stable in most cases. **B** illustrates the course of ctDNA AFs of two patients responding to combined ICI.

For patient 23, the variant AFs quickly decrease and remain below the detection limit. **C** shows two patients with adjuvant ICI and a relapse during follow-up. The treatment of patient 74 was changed upon progression and the clinical response to the new treatment is reflected in decreasing AFs below the detection limit. For patient 85, we detected progression under adjuvant treatment at day 43 (64 days before PET/CT). All panels: Each line represents a single somatic variant before, during or after treatment. Source data are provided as a Source Data file.

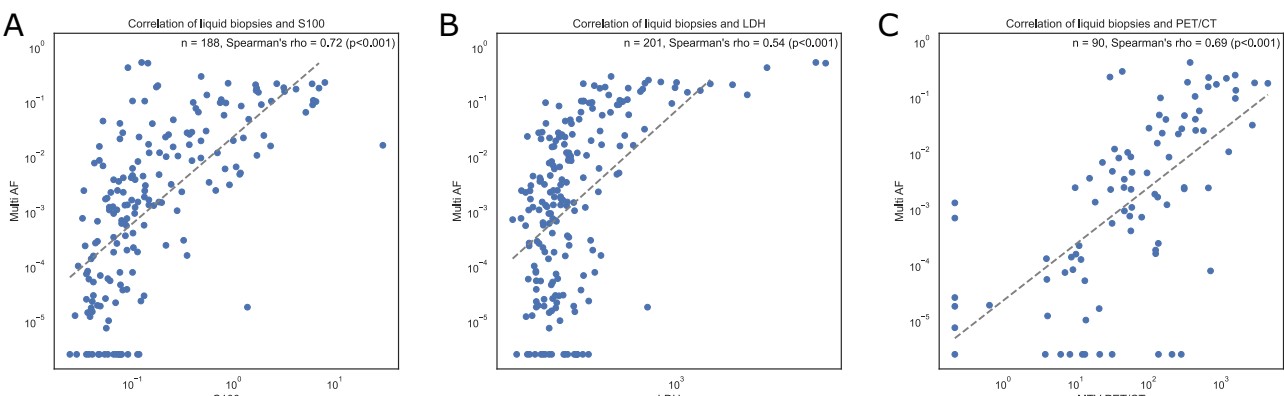

**Fig. 3 | Correlation of mean AFs and known melanoma biomarkers. A**, **B** and **C** show the mean allele fraction of patient-specific variants in liquid biopsy samples from patients under palliative combined ICI and S100 (µg/l), LDH (U/l) and MTV (ml). The Spearman's rank correlation coefficients were calculated to determine the correlation between AF and S100 (rho = 0.72, $t$ = 8.17, df = 62, $p$ = 2.01 × 10⁻⁹), LDH (rho = 0.54, $t$ = 5.06, df = 62, $p$ = 4.12 × 10⁻⁴) and MTV (rho = 0.69, $t$ = 7.26, df = 58, $p$ = 1.1 × 10⁻⁶). Axes were log transformed and the minimum of all non-zero values was divided by two and added to zero values. S100 - µg/l, LDH - U/l, MTV - ml. Source data are provided as a Source Data file.

disease recurrence in patients with advanced melanoma treated with combined or adjuvant ICI.

Detection of cell-free DNA is a minimally invasive blood test that allows repeated sampling. The approach used in this study combines a larger number of patient-individual tumour-specific mutations

resulting in a greater flexibility compared to single or few hotspot mutations[28–31]. This is particularly important for *BRAF* or *NRAS* wildtype patients. In our cohort typical hotspot mutations in *BRAF* or *NRAS*, were found in only 72% of patients. We included up to 30 independent somatic mutation loci per patient, which were each sequenced to an

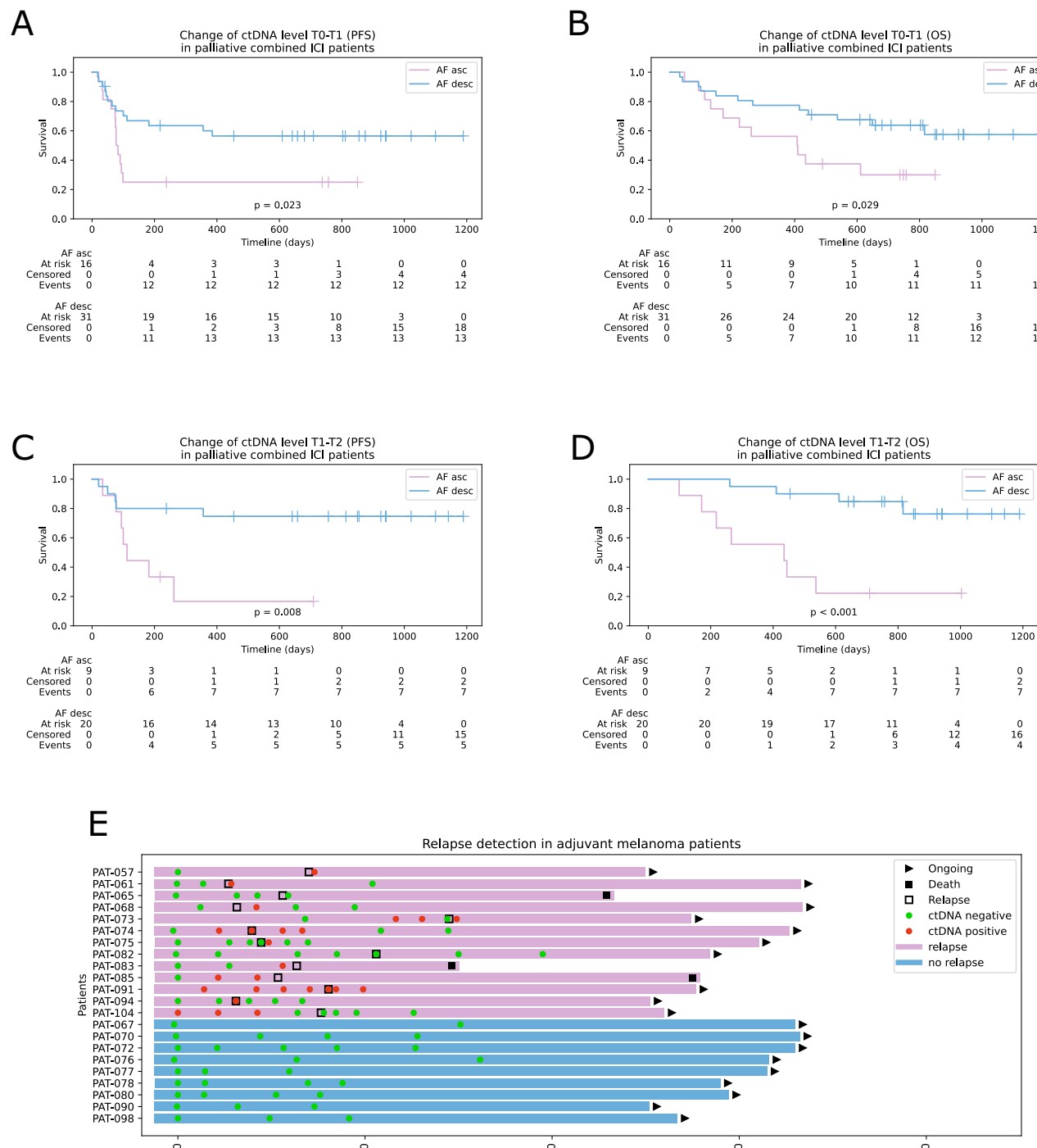

**Fig. 4 | Survival analysis in advanced melanoma patients and relapse detection in adjuvant patients.** Kaplan-Meier curves and differences in survival for patients with combined ICI grouped by ctDNA AF changes between T0 and T1 for PFS (**A**), OS (**B**) and between T1 and T2 for PFS (**C**) and OS ($p = 3.712 \times 10^{-4}$) (**D**). AF asc− ascending or stable allele fractions over time, AF desc−descending allele fractions.

**E** A swimmer plot with patients with adjuvant ICI and follow-up. Relapse-free patients (blue bars) had no ctDNA-positive liquid biopsy indicating a high specificity of the approach. ctDNA-negative: samples with less than three significant variants, ctDNA-positive: samples with at least three significant variants. Source data are provided as a Source Data file.

average depth of ~6.000× after deduplication. Only reads with at least one duplicate were kept for subsequent analysis, requiring more than 50,000× raw coverage. The combination of ultra-deep sequencing and UMIs allowed the reduction of error rates to below 0.1% and even lower for higher duplication levels (e.g., <0.01% for reads with 4 copies). The detection rate of tumour variants at T0 in this study was 87% of advanced melanoma patients, similar to 90% in a previous series of 29

patients using a different PCR-based tumour-informed approach with 16 mutations[32].

The prognostic relevance of LDH, TMB and PET/CT in patients with advanced melanoma was confirmed in our cohort[33,34]. Furthermore, we show that allele fractions in plasma correlated with LDH, S100 and MTV which is consistent with studies using e.g. digital droplet PCR for ctDNA detection in similar patients[15,31,35]. Interestingly, the

correlation of allele fractions was higher for S100 compared to LDH. While PET is the current gold standard for assessing metabolic tumour volume, we found a rare case of a patient with metabolic negative melanoma metastases. This illustrates that ctDNA and PET provide complementary information as disease progression was detected by ctDNA and CT, but not by PET.

While ctDNA at T0 was not related to OS in patients with advanced melanoma, the dynamic of ctDNA at the beginning of treatment (after three weeks of treatment) was highly predictive for OS and PFS. Decreasing ctDNA between T0 and T1 was associated with better prognosis, which is in line with previous work by us and others[15,32,35,36]. In addition, the current study showed an even stronger prognostic value for ctDNA changes between T1 and T2. This could be relevant in routine clinical practice, as monitoring with ctDNA can still be started after the initiation of ICI to predict treatment response, e.g. in patients with side effects during therapy. If the tumour-informed panel is designed at the time of the start of ICI treatment, sequencing of subsequent liquid biopsies is possible within a few days after arrival of the samples. The results could therefore be used directly in clinical practice. Besides the predictive and prognostic value of ctDNA in patients under ICI, measuring ctDNA levels during the follow-up of adjuvant melanoma patients can be used to detect relapse up to several months before clinically apparent recurrence[32]. In our cohort of adjuvant patients, we were able to detect ctDNA in 76.9% of patients up to 133 days before clinical progression. Early detection of recurrence helps to avoid keeping patients on futile treatment. Discontinuation of treatment may prevent immune-mediated side effects and allows to save costs.

Finally, the tumour-informed cell-free DNA test used in this study allows the generation of results in a diagnostic relevant time frame. A patient-specific panel can be designed and delivered in as little as one week. Once the enrichment panel is available, liquid biopsies can be sequenced and analysed within two weeks. To date, consensus quality parameters (e.g. limit of detection, threshold for minimum allele fractions, minimum coverage of unique molecules per monitored variant, error correction by UMIs, sample quality control) for tumour-informed liquid biopsies are lacking. National and international efforts are underway to define standards and requirements for this type of testing for broader clinical use.

There are limitations of the current study, mainly the cohort size and the timing of liquid biopsy collection during routine clinical care. Differences in ctDNA levels and biomarkers as well as imaging results could also be explained to some extent by the different sampling times. Nevertheless, we believe this study helps to design future studies using tumour-informed liquid biopsies in melanoma patients.

In summary, the results of this study confirm the prognostic relevance of ctDNA in patients with advanced melanoma treated with ICI. Measurement of ctDNA levels in plasma using multiple patient-specific mutations provides more information than S100 and LDH, and is complementary to state-of-the-art imaging with PET/CT. In addition, tumour-informed liquid biopsies are a novel approach in melanoma patients that comprises far more than the hotspot mutations and allows the detection of patient-specific tumour variants with higher sensitivity, which could be even further improved by increasing the number of monitored variants (e.g. from 30 to 100). Most importantly, our approach allows the design of tumour-informed panels within one week, which enables prospective use in diagnostics. To this end, whole exome sequencing of tumour-normal pairs will provide substantially more somatic variants for monitoring compared to the 700+-gene panel used in this study. Finally, the addition of resistance markers could enhance clinical utility, including early detection of emerging therapeutic resistance.

## Methods
### Patient cohort
All patients included in this study gave their written consent and the study was approved by the local ethical review board (project ID: 196/

2019BO2). We included melanoma patients who started combined ICI with ipilimumab and nivolumab between July 2019 and May 2021 of whom tumour tissue was available. From September 2020 onwards, adjuvant ICI patients were also included via an amendment in order to evaluate ctDNA also as biomarker for the detection of relapse. After successful NGS of the metastatic tissue and somatic variant detection, liquid biopsies were taken at the regular laboratory checks before and during ICI in order to monitor all driver and selected passenger mutations of the tumour (up to 30 mutations in total per patient, Supplementary Fig. 3) during ICI. As the combined ICI is administered at intervals of three weeks, a liquid biopsy was usually collected every 3 weeks, as well as at the time of PET/CT, which was usually done at baseline and 12 weeks after the start of ICI. The final analysis included samples before the first cycle of ICI, that is baseline (T0, corresponding to ≤7 days of ICI start), at the time of the second cycle of ICI after 3 weeks (T1, corresponding to ±7 days of the second course of ICI) and at the time of the first follow-up staging with PET/CT (T2, corresponding to ±21 days of the second PET/CT), as well as additional time points in a subset of patients. For the cohort of adjuvant patients, no specific time points were defined and plasma samples were collected at every ICI cycle and analysed for early detection of relapse.

Tumour-normal sequencing and data analysis were performed on DNA from FFPE material[37]. Tumour-DNA was isolated following standard protocols from FFPE blocks. The samples were macro-dissected from one to ten 5 μm paraffin sections depending on tumour size. DNA was extracted using the Maxwell RSC DNA FFPE Kit and the Maxwell RSC Instrument (Promega, Madison, WI, USA) according to the manufacturer's instructions. There was no pre-defined threshold for histological tumour content. Samples were excluded if there was no tumour left on the block. Tumour and normal DNA were sequenced using a 700+ custom gene panel and analysed using the megSAP NGS analysis pipeline (https://github.com/imgag/megSAP). The resulting annotated variants were visualised and further investigated (e.g. inspection of read alignment, identification of driver mutations based on public databases) using the clinical decision support system GSvar (https://github.com/imgag/ngs-bits). Variants were classified according to the VICC guidelines[38].

### Sequencing of cell-free DNA
Specific vacutainers were used to collect blood samples (Streck, La Vista, USA). Upon arrival in the lab the blood samples were centrifuged twice to separate plasma from cells. Afterwards, the plasma was stored at −80 °C until further processing. Isolation of cell-free DNA from plasma was done with the QIAamp MinElute ccfDNA Kits (Qiagen, Hilden, Germany). The quality and fragment size distribution of cell-free DNA was analysed on a Fragment Analyzer using the High Sensitivity Large Fragment Analysis kit (Agilent, Santa Clara, US). Cell-free DNA was quantified using the Qubit dsDNA HS Assay kit (Thermo Fisher Scientific, Waltham, USA). The xGenPrism DNA Library Prep Kit (now renamed to xGen cfDNA & FFPE DNA Library Preparation Kit, IDT, Coralville, USA) was used to construct libraries from on average 37 ng of cell-free DNA (range 1–84 ng). The protocol includes a barcoding step in which fixed single-stranded unique molecular identifier (UMI) sequences are added to each cell-free DNA molecule before amplification. The target regions were enriched using hybridisation capture (xGen Hybridisation and Wash Kit, IDT, Coralville, USA). Up to 30 patient-specific somatic mutations were selected for the design and synthesis of 5'-biotinilatyed oligo probes for the tumour-informed capture panels (NGS Discovery Pool, IDT, Coralville, USA). An individual panel was designed and ordered per patient, which, in addition to the patient's selected somatic mutations, also included a set of germline variants for patient identification, quality control and statistics. The selection of mutations was based on a set of selection criteria, e.g. high-quality predictions, mutations with a high clonality (minor allele fraction) and driver mutations preferred. During analysis,

variants in homopolymers and other repetitive contexts, variants outside the target region of the tumour sequencing and variants with high frequency in public databases such as gnomAD were removed. In addition, fingerprint SNVs (germline SNVs used for sample identification) were included in the panel. A list of tumour variants used to monitor ctDNA can be found in the supplement (Supplementary Data 1). Subsequent sequencing with an intended depth of 100,000× was performed on a NovaSeq6000 (Illumina, San Diego, USA).

Sequencing error correction and detection of low-frequency variants was done using UMIs. We developed the analysis tool umiVar (version 2023_06_2, https://doi.org/10.5281/zenodo.12755789), to enable efficient sequencing error correction and accurate detection of ultra-low fractions of ctDNA using UMI-barcoded reads. UmiVar is integrated in the megSAP platform (version 2022_08-173-gdc428fb4, https://github.com/imgag/megSAP). In brief, after demultiplexing an adapter-trimming was performed on the FastQ files[39], the reads were mapped using standard bwa-mem2 (version 2.2.1, https://github.com/bwa-mem2/bwa-mem2) mapping (-K 100000000 -Y) following an indel-realignment using abra2[40], all reads were grouped by their UMI and mapping position using the barcode correction script of umiVar, revealing copies of the same cell-free DNA molecule. Several bioinformatic data quality checks were done, which included fragment size (Supplementary Fig. 8), sequencing depth, and correlation of SNP fingerprints. Next, umiVar uses duplicates for error correction by consensus generation, with increasing error-correction efficiency the more copies of a molecule are available (termed 2-fold, 3-fold and 4-fold error correction if 2, 3, or >=4 copies of a molecule were sequenced and used for consensus generation, respectively). Reads without duplicates were excluded from further analysis.

umiVar generates separate probabilistic error models based on a beta-binomial distribution of errors for each possible nucleotide change and error-correction level (i.e. 2-fold, 3-fold and 4-fold error correction). By integrating reads of all error-correction levels for a specific monitored variant, umiVar then generates an alternative allele count (altC), minor allele fraction (MAF), a $p$ value (probability that the observed variation is not present in the sample) and false discovery rate for each monitored variant and the sample. Variants with a $p$ value < 0.05 were considered as detected. Furthermore, umiVar allows to interrogate each error-correction level separately, i.e., using only >= 4-fold corrected reads for improved precision in case of very high read-depth (Supplementary Table 1).

Making use of the added information from up to 30 monitored variants and multiple time points (samples) per patient, we applied additional quality filters removing variant positions with low sequencing depth (coverage <1000× with reads of 2-fold or higher correction level), small insertions and deletions and variants with a MAF more than 3 standard deviations higher than the mean MAF of all variants at the respective time point (outlier removal). The MAFs of the remaining (high quality) variant positions were used to generate monitoring curves along the sampled time points that visualise changes in ctDNA levels during treatment.

### Imaging and image analysis

All PET/CT examinations were performed in-house according to a standardised acquisition protocol on a single clinical scanner (Siemens Biograph mCT, Siemens Healthineers, Knoxville, USA) following international guidelines.

Diagnostic whole-body CT was acquired in expiration with arms elevated according to a standardised protocol using the following scan parameters: reference tube current exposure time product, 200 mAs with automated exposure control (CareDose); tube voltage, 120 kV. CT examinations were performed with weight-adapted 90–120 ml intravenous CT contrast agent in a portal-venous phase (Ultravist 370, Bayer Healthcare) or without contrast agent (in case of existing contraindications). CT data were reconstructed in transverse orientation

with a slice thickness between 2.0 mm and 3.0 mm with an in-plane voxel edge length between 0.7 and 1.0 mm[18]. F-FDG was injected intravenously after at least 6 h of fasting. PET acquisition was initiated 60 min after injection of a weight-adapted dose of approximately 300 MBq 18F-FDG (314.7 MBq ± 22.1 MBq). PET was acquired over four to eight bed positions (usually from the skull base to the mid-thigh level) and reconstructed using a 3D-ordered subset expectation maximisation algorithm (two iterations, 21 subsets, Gaussian filter 2.0 mm, matrix size 400 × 400, slice thickness 3.0 mm, voxel size of 2.04 × 2.04 × 3 mm³). PET acquisition time was 2 min per bed position.

A deep learning approach was used for extraction of quantitative measures of tumour burden and tumour metabolism from PET/CT image data. To this end, a publicly available, pretrained neural network, based on the nnUNet framework was deployed for automated segmentation of metabolically active tumour lesions on FDG-PET volumes[41–43]. MTV and TLG were computed from obtained segmentation masks[44].

### Data integration and combined analysis

Data from routine diagnostic work-up such as the biomarkers S100 and LDH as well as results of tumour-normal sequencing, including tumour mutation burden, were extracted from electronic health records with custom scripts. Survival analysis and correlations were done using python and the packages lifelines 0.27.3, pandas 1.4.0, matplotlib 3.5.1, seaborn 0.11.2 and scipy 1.7.3. Samples with a residual tumour $p$-value of <0.05 in umiVar were considered ctDNA-positive. Samples with three analysable variants were included in correlation analyses and for the relapse detection. In the adjuvant cohort, a liquid biopsy was considered ctDNA-positive if ≥3 tumour variants were detected with a $p < 0.05$. Correlation analyses were done for allele fractions, tumour TLG and MTV. For the survival analysis, increasing allele fractions were defined as $AF(t_n - t_{(n+1)}) \geq 0$ and decreasing allele fractions as $AF(t_n - t_{(n+1)}) < 0$.

### Statistics and reproducibility

No statistical method was used to predetermine sample size. Instead, in this exploratory study, all melanoma patients receiving ICI between 2019 and 2021 were offered to participate. Patients or samples have been excluded based on several quality control parameters (DNA quality, sequencing data quality, minimum number of somatic variants for monitoring) as described in the manuscript. Randomisation was not applicable. The study was unblinded. No correction for multiple testing was applied due to the exploratory nature of the study. Spearman correlation was selected because of the non-linearity after visual inspection of scatterplots. $P$ values for Spearman correlations (rho) were determined using the t-transformation of rho with degrees of freedom $n - 2$ and n equal to the number of patients, not the number of measurements, because multiple samples of one patient are not independent. The level of significance was 0.05 (two-sided).

### Reporting summary

Further information on research design is available in the Nature Portfolio Reporting Summary linked to this article.

## Data availability

The raw sequencing data generated in this study have been deposited in the GHGA database under accession code GHGAS40406834890492. The raw data are available under restricted access for data privacy reasons and access will be granted by the data access committee (DAC) of this project. Requests for access can be sent via the GHGA portal or to the corresponding author directly. The access will be restricted to requests from authenticated users and assessed by the DAC, the local institutional review board and the local data security office. The estimated timescale for data access is three months. Processed data are available on Github (https://github.com/imgag/PET_LIT_study). Data

used to create tables and figures can be found in the Source Data file provided with this paper.

## Code availability
The umiVar tool is available on Github (https://github.com/imgag/umiVar). The umiVar version used to analyse the data in this study has been archived on Zenodo (https://doi.org/10.5281/zenodo.12755789). The NGS platform megSAP and ngs-bits are available on Github (https://github.com/imgag/megSAP, https://github.com/imgag/ngs-bits). Additional scripts, used for data analysis and the creation figures and tables are available in a separate Github repository (https://github.com/imgag/PET_LIT_study).

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

## Acknowledgements

We thank all the patients and their families for participating in this study. This study was supported by a research grant of the Bristol Myers Squibb Stiftung Immunonkologie (project FA 19-008). S.O. and C.S. are supported by funding from the European Union's EU4Health programme under grant agreement No 101080009 (Can.Heal) and the German Consortium for Translational Cancer Research (DKTK) Joint Funding programme (EXLIQUID).

## Author contributions

A.F., C.P., B.G., S.O., C.S. designed the study. O.K., S.G., I.B. and T.K. performed the experiments. F.M., L.S., S.O. designed and implemented the analysis tools. A.F. collected samples and clinical data, S.G. performed evaluation of radiologic results, L.S., J.A., P.M., S.A.E., S.G. and C.S. assembled the input data and analysed the data. S.O., C.P., C.G., L.F., P.M., S.A.E., S.G. and A.F. supervised the analysis. S.O., A.F., C.S. prepared the manuscript. All authors edited the manuscript.

## Funding

## Competing interests

A.F. reports honoraria for presentations for BMS, MSD, Novartis, Pierre-Fabre, Delcath and Immunocore; travel support and congress participation support from BMS, Pierre-Fabre, Novartis, MSD; Advisory Boards from MSD, BMS, Novartis, Pierre-Fabre, Immunocore and research funding from BMS Stiftung Immunonkologie. C.S. and S.O. report research funding from BMS Stiftung Immunonkologie as well as institutional grants and payment for conference presentations from Illumina Inc. and Oxford Nanopore Technologies outside the submitted work. The remaining authors have no conflict of interests to disclose.
