## [Peer Review File · Nature Communications]

Tumour-informed liquid biopsies to monitor advanced melanoma patients under immune checkpoint inhibitionREVIEWER COMMENTS

Reviewer #1 (Remarks to the Author): expertise in ctDNA methods

The authors reported detection of ctDNA in patients receiving adjuvant ICI using a personalized panel of tumor-informed genes. While the cohort is small, the increased number of targets may improve detection of relapse.

However, more information is needed in order to determine the strength of the authors conclusions and be acceptable for publication. The following changes are suggested and would provide more clarity to the study.

Results & Methods:

- Is it stated that 15% of samples were removed because of quality issues. Does haemolysis affect the assay? Are genomic fragments not removed with the second spin of the Streck tubes? Has the assay been clinically validated? Does the quality control check for large fragments (in other words, genomic DNA)? Was this the reason some samples were excluded?

Results:

- Please include the stages of patients analyzed. The recurrence rate is quite high.

- Need correction of typos. example: second paragraph of "Survival Analysis" this is Figure 4E, not 2E as stated.

Discussion:

- When stating a detection rate of 87% at T0 the authors need to include a comparison with non-NGS methods of ctDNA detection. example: Syeda et al (Lancet Onc 2021) demonstrated a detection rate of 93% at baseline using ddPCR with a single tumor informed hotspot mutation. This was a higher detection rate in unresectable patients. However, it is unclear if both publications are referring to the same target population.

- For the detection of ctDNA in the adjuvant setting the authors state a detection rate of 53.8%. This is a high detection rate for adjuvant setting, given that the term "adjuvant" may now apply to stages IIB to IV, which has a wide spread in terms of relapse rates. An ASCO report on Checkmate-915 by Georgina Long reported a 16% ctDNA positivity rate at baseline in patients with IIB-D/IV resected melanoma receiving ICI. Additionally, the authors statement does not indicate if this is the detection rate at baseline (T0) or at any positive ctDNA measurement (including at the time of progression). Please specify the clinical stage of this set of patients.

- There seems to only be 6 patients that were detected before clinical relapse and 4 patients were detected upon clinical relapse or after. The authors claim that this may be used to end

treatment early, the evidence does not support this. However, the assay is very specific and none of the non-relapse patients had a false positive.

-In the limitations section one of the limitations may also be the amount of DNA needed to analyze the samples. Were all the samples required to have 200ng of DNA as stated in reference #33? This may not be an issue for unresectable melanomas but may become difficult in the adjuvant setting.

Methods:

- Please include patient clinical stage and how many in each category.

Additionally, the nomenclature of Group 1 and Group 2 is not used in any other part of the manuscript and may not be needed.

- How many slides were used? Was there a threshold for tumor content 10%?

- What was the least number of mutations used? Can you provide the range and average of mutations used in the cohort?

Figure 1:

- A range of timeline for T2 would be nice to have in this figure

Figure 2:

_ Make a distinction between molecular progression and clinical progression. For patient 74, was treatment changed based on clinical progression? If so, what was the means of detecting it?

Figure 4:

In the timeline, specify the unit. Days?

Table:

- Please add more information to Table 1. Is it possible to add stage (III vs IV)? Perhaps patient status at last follow up (dead vs alive)? Median tumor thickness and range? Ulceration present vs absent?

Reviewer #2 (Remarks to the Author): clinical expertise in melanoma immune checkpoint inhibitor usage

The authors conducted a very nice study (prospective) on 87 melanoma patients receiving combination regimen ICI (65 patients in the palliative setting and 22 patients as adjuvant therapy) for monitoring the ctDNA with an approach already described (publications mentioned in the references) but not yet used in prospective manner for monitoring relapse.

From my perspective the fact of using this method (patient-specific tumor variants) is not enough innovative for publication in Nature Communication (impact factor 16.6); the concept of using the ctDNA for monitoring is already a clear concept and the method used in this study already published.

The study is very well conducted and very well written; definitely worth published but not in this journal from my opinion.

You mentioned that BRAF or NRAS, were found in less than 12% of your patients - why is this?

The majority of BRAF went for BRAF/MEK inhibition and not for ICI?

What about TERT, CDKN2A, RAC1, PTEN, NF1?

Can you give more details concerning the variants analysed? Are there some that are common in a high percentage of patients? Or the majority are really patient specific (different variants for each patient)?

Can your method can be simplified to be used in clinical practice? At this point is it possible to be used as it is? How much time is needed from blood collection /tumor collection till having the 30 variants analysed in the blood?

Reviewer #3 (Remarks to the Author): expertise in immunotherapy clinical trials

well written and well conducted study

comments:

1. The gold standard in imagin is still CT rather than PET/CT. I recognise there has been more data recently supporting the use of PET/CT in assessing response in melanoma patients and also hypothesising that its use can be useful in better determining response and when a patient may stop therapy within the context of toxicity/other issues with treatment. I think a breif reason why PET/CT was used here as opposed to CT alone would be useful, maybe split ebetween introduction and discussion as appropriate. There is no or little discussion around the issues of response assesmsnet such as pseudoprogression and also stable disease on CT and how this relates in reality to biological response and long term outcome.

2. Can the authors comment briefly on how applicable their technique is for use now and how the field may develop; standardisation of methodology and how, briefly comment on how comparable techniques are and if there is an evolving common concensus on what constitutes a positive or negative result. The patient graphs they show in figure 2 show that the levels in 027 did go down but then icncreased again and not to the level of baseline. Is it going to be possible to define categories of responses or is this always going to be an individual response with several readings of ctDNA per patient to make the assessment?

3. it is intriguing to see the palliative and adjuvant graphs in figure 2 as indicating that ctDNA may be used to tailor treatment such as using single agent ICI then combination if toxicity may be an issue. Such a trial design may be possible along with the genetic markers of IO txocitiy or

immune cell monitoring eg Bcell work that has been published over the past 2-3 years indicating that these subcells can identify patients prone to IO toxicity.

4. I found it hard to follow the use of cfDNA here and ctDNA when they have essentially only looked at ctDNA. is it necessarily to use both terms here?

RESPONSE TO REVIEWERS' COMMENTS

We would like to thank the reviewers and you for the careful and thorough assessment of the manuscript. We have addressed all points raised by the referees and hope that the responses are satisfactory. With these revisions, we believe that our manuscript has been substantially strengthened and hope that it is now suitable for publication in Nature Communications.

Please find below our point-by-point replies to the reviewers' comments. All changes in the main manuscript and the supplement have been marked in red. We refer to the changes we have made in the manuscript using consecutive line numbering.

Reviewer #1 (Remarks to the Author): expertise in ctDNA methods

The authors reported detection of ctDNA in patients receiving adjuvant ICI using a personalized panel of tumor-informed genes. While the cohort is small, the increased number of targets may improve detection of relapse.

However, more information is needed in order to determine the strength of the authors conclusions and be acceptable for publication. The following changes are suggested and would provide more clarity to the study.

Answer: We thank the reviewer for the constructive criticism and many interesting suggestions, which led to new analyses and results that we have now added to the manuscript. We would like to add that the overall cohort with more than 100 patients and examinations at multiple time points is comparable to other recent studies in lung cancer using tumour-informed liquid biopsies (e.g. PMIDs 37683638, 34505064).

Results & Methods:

- Is it stated that 15% of samples were removed because of quality issues. Does haemolysis affect the assay? Are genomic fragments not removed with the second spin of the Streck tubes? Has the assay been clinically validated? Does the quality control check for large fragments (in other words, genomic DNA)? Was this the reason some samples were excluded?

Answer: Fifty-five samples were excluded in total (s. filter criteria in the methods section starting at line 367 and results section starting at line 122). Four samples were haemolytic, indicating that the collection with Streck tubes worked well overall. We decided to exclude the haemolytic samples as there is evidence that the amount of genomic (non-tumorous) DNA in plasma samples increases with haemolysis and therefore the fraction of cell-free tumour DNA could be underestimated (PMIDs 31639669, 31639669). The second spin is intended to reduce the contamination with cells from the buffy coat, again increasing the ctDNA fraction while avoiding additional cell-free normal DNA (PMID 30094369). The samples were not fragmented during library preparation. Long genomic fragments are detected by quality control, in this case the Fragment Analyzer and Qubit (s. methods section line 325), and we did not observe a peak beyond the expected fragment length of cell-free DNA.

Bioinformatically, we always compute the fragment (insert) size distribution, but did not observe significant amounts of long fragments, apart from small peaks at 320 bp (2 histone windings) and 480 bp (3 histone windings) (new supplementary figure 5), which are expected for cell-free DNA. If observed, fragments with unexpected insert sizes can be removed bioinformatically. This is a parameter in our software.

Somatic variants in white blood cells (potential source of genomic DNA) are mostly found in a few well-known genes associated with clonal haematopoiesis of indeterminate potential (CHIP) such as *DNMT3* and *TET2*. These genes could be excluded from tumour-informed panels to further reduce the chance of mis-interpreting CHIP variants as tumour variants. However, the chance that a CHIP variant in white blood cells occurs in one of the 30 monitored SNV positions is extremely low and can be neglected. CHIP mutations are mainly an issue for liquid biopsy methods using large gene panels, exomes or even genomes. We now discuss this topic briefly in the legend of the new supplementary figure 5.

Finally, our assay has been validated using an established ctDNA standard (Twist cfDNA Pan-Cancer Reference, Twist Bioscience, San Francisco, US) showing the sensitivity and precision required for this study.

Results:

- Please include the stages of patients analyzed. The recurrence rate is quite high.

Answer: We added the disease stage to the manuscript starting at line 124. Details of the final cohort can be found in table 1.

- Need correction of typos. example: second paragraph of "Survival Analysis" this is Figure 4E, not 2E as stated.

Answer: We thank the reviewer for this comment and apologize for the wrong figure reference which, in addition to a smaller number of other typos, were corrected in our revised version of the manuscript. The changes have been highlighted in the manuscript.

Discussion:

- When stating a detection rate of 87% at T0 the authors need to include a comparison with non-NGS methods of ctDNA detection. example: Syeda et al (Lancet Onc 2021) demonstrated a detection rate of 93% at baseline using ddPCR with a single tumour-informed hotspot mutation. This was a higher detection rate in unresectable patients. However, it is unclear if both publications are referring to the same target population.

Answer: In our palliative cohort, as well as in the cohort of Syeda, only patient with non-resectable metastases were included. In our cohort, 100% of the patients were at stage IV, in the study of Syeda 96% of the patients were at stage IV and 4% unresectable stage III. Syeda and colleagues evaluated samples of the two cohorts COMBI-d and COMBI-MB, and therefore only patients with *BRAF* V600 mutation were included in their analysis, while *BRAF* V600 wildtype patients could not take part in these studies. Our study included both, *BRAF*-mutated and -wildtype patients. The fraction of patients with *BRAF* V600 mutation was 29%. Another major difference is that patients with previous systemic treatment for advanced or metastatic melanoma had been excluded from COMBI-d and COMBI-MB. In our cohort, only 43% of the patients received ICI as first-line treatment in the palliative cohort and therefore the majority of patients had a previous systemic treatment. In summary, these substantial differences result in a limited comparability between the two studies.

- For the detection of ctDNA in the adjuvant setting the authors state a detection rate of 53.8%. This is a high detection rate for adjuvant setting, give that the term "adjuvant" may

now apply to stages IIB to IV, which has a wide spread in terms of relapse rates. An ASCO report on Checkmate-915 by Georgina Long reported a 16% ctDNA positivity rate at baseline in patients with IIIB-D/IV resected melanoma receiving ICI.

Additionally, the authors statement does not indicate if this is the detection rate at baseline (T0) or at any positive ctDNA measurement (including at the time of progression). Please specify the clinical stage of this set of patients.

Answer: We want to thank the reviewer for his comment. The Checkmate-915 study analysed post-resection pre-treatment plasma samples (DOI: 10.1016/j.annonc.2022.07.914). The detection of ctDNA-levels only in pre-treatment samples is an important difference to our study as we also report the detection rate of relapse by ctDNA. These data are not reported by Gergina Long and colleagues.

- There seems to only be 6 patients that were detected before clinical relapse and 4 patients were detected upon clinical relapse or after. The authors claim that this may be used to end treatment early, the evidence does not support this. However, the assay is very specific and none of the non-relapse patients had a false positive.

Answer: The detection of relapse was hampered for few of our patients by missing blood samples at the exact time of clinical relapse. This is a limitation regarding the clinical conclusions and mentioned in the limitations section. With this in mind, we found five of thirteen patients with clinical progress ctDNA-positive at the time of progress in the adjuvant cohort (Pat. PAT-074, PAT-083, PAT-085, PAT-091, PAT-094). Three additional patients were tested positive close to the progress (PAT-057, PAT-061, PAT-068), two of them within 7 days. Interestingly, three more patients were ctDNA-negative at clinical progress but were either ctDNA-positive in additional liquid biopsies before or thereafter (PAT-073, PAT-075, PAT-104). Although we have to admit that the numbers are small and the findings have to be validated in a larger clinical trial, we believe that our test can be used to efficiently detect relapse.

-In the limitations section one of the limitations may also be the amount of DNA needed to analyze the samples. Were all the samples required to have 200ng of DNA as stated in reference #33? This may not be an issue for unresectable melanomas but may become difficult in the adjuvant setting.

Answer: During library preparation, we used on average 37 ng of cell-free DNA (range 1-84 ng, n = 321). This is an importation parameter and we thank the reviewer for bringing this up. We added the information to the manuscript in the methods section. As a side comment, the average amount of cell-free DNA used in the subset of adjuvant patients was slightly lower (35 ng, range 5-66 ng, n = 98). We assume that the amount of 200 ng DNA referred to by the reviewer is based on the tumour-normal FFPE sequencing which was done as described in our previous study (PMID 35192052). The diagnostic Tumour-normal sequencing is following a different protocol and is only used to identify somatic variants for therapy/relapse monitoring.

Methods:

- Please include patient clinical stage and how many in each category. Additionally, the nomenclature of Group 1 and Group 2 is not used in any other part of the manuscript and may not be needed.

Answer: The clinical stage was added to the manuscript in table 1. We removed the labels Group 1 and Group 2 in the methods section.

- How many slides were used? Was there a threshold for tumor content 10%?

Answer: We added the information on the number of slides to the methods section (s. methods section starting at line 310). There was no threshold for histological tumour content. Samples were removed if no tumour was detected histologically.

- What was the least number of mutations used? Can you provide the range and average of mutations used in the cohort?

Answer: For the majority of samples, we designed tumour-informed panel with 30 somatic variants (average number was 22, median = 30, range from 3 to 30 variants, see new supplementary figure 2 for a histogram). Patients with smaller tumour-informed panels did not have 30 somatic SNVs or indels to choose from. One of the limitations of this study was the use of a large panel (700+ genes) for tumour sequencing, which limits the number of detectable somatic mutations. Exome sequencing would allow to increase the number of variants and most likely the sensitivity of the assay. We have therefore very recently switched to exome-sequencing of tumours for patients of our molecular tumour board.

Figure 1:

- A range of timeline for T2 would be nice to have in this figure

Answer: We added a timeline to figure 1, but found it overloaded. The timeline can now be found in the methods section (s. patients and methods section starting at line 300).

Figure 2:

_ Make a distinction between molecular progression and clinical progression. For patient 74, was treatment changed based on clinical progression? If so, what was the means of detecting it?

Answer: Therapy monitoring and treatment intervention upon detection of progression is one of most promising applications for ctDNA. Our study was designed as an observational trial and did therefore not allow treatment changes based on ctDNA. We are currently preparing follow-up studies that allow treatment interventions based on changes of ctDNA levels. The disease progression in patient 74 would have been detected much earlier with the ctDNA results.

Figure 4:

In the timeline, specify the unit. Days?

Answer: Yes, the unit is days. We have added this information to the timeline in figure 4.

Table:

- Please add more information to Table 1. Is it possible to add stage (III vs IV)? Perhaps patient status at last follow up (dead vs alive)? Median tumor thickness and range? Ulceration present vs absent?

Answer: As recommended by the reviewer, we added additional clinical information to table 1. The recommended parameters ulceration and tumour thickness are part of the AJCC classification and included there.

Reviewer #2 (Remarks to the Author): clinical expertise in melanoma immune checkpoint inhibitor usage

The authors conducted a very nice study (prospective) on 87 melanoma patients receiving combination regimen ICI (65 patients in the palliative setting and 22 patients as adjuvant therapy) for monitoring the ctDNA with an approach already described (publications mentioned in the references) but not yet used in prospective manner for monitoring relapse. From my perspective the fact of using this method (patient-specific tumor variants) is not enough innovative for publication in Nature Communication (impact factor 16.6); the concept of using the ctDNA for monitoring is already a clear concept and the method used in this study already published. The study is very well conducted and very well written; definitely worth published but not in this journal from my opinion.

Answer: We thank the reviewer for the constructive criticism and critical evaluation of our method. The protocols currently used to analyse cell-free DNA are diverse and currently not well standardized. We are aware of two studies that used tumour-informed panel designs for liquid biopsy in melanoma patients. One had a large cohort of patients with a single pre-treatment sample, i.e. did not address the problem of treatment monitoring (CHECKMATE-915 in melanoma DOI: 10.1016/j.annonc.2022.07.914), and a second study which used a far smaller number of monitoring variants (PMID 36869646).

In addition, the relapse detection in the adjuvant setting using tumour-informed hybridization panels as well as the comparison with PET/CT has not been shown before. Also, our method has not been published. In previous studies we have used very small number of variants (1-5) while here we use up to 30 variants and can support even higher variant numbers. Likewise, the umiVar software has not been published before.

Most importantly, our new approach facilitates the design of tumour-informed panels within one week, which enables prospective use in diagnostics, which was not possible with the method described in our previous paper (PMID 32687856). We added the discussion about the diagnostic usability of the method to the manuscript (line 259)

You mentioned that BRAF or NRAS, were found in less than 12% of your patients - why is this? The majority of BRAF went for BRAF/MEK inhibition and not for ICI?

Answer: We thank the reviewer for this comment. This was an error in our manuscript which we corrected. The correct fraction of patients with hotspot mutations in *BRAF* and/or *NRAS* is 64%. We changed the manuscript accordingly and added an additional figure summarizing all (i.e. not only hotspot) *BRAF* and *NRAS* mutations as well as mutations in other driver genes to the supplement (supplementary figure 1).

What about TERT, CDKN2A, RAC1, PTEN, NF1?

Can you give more details concerning the variants analysed? Are there some that are common in a high percentage of patients? Or the majority are really patient specific (different variants for each patient)?

Answer: The variants for monitoring were selected based on an algorithm described in the methods section. The algorithm selects up to 30 mutations, prioritising known drivers, somatic variants with high allele fraction (more likely to be detectable in plasma), as well as high quality variants. We also prioritise SNVs over indels due to higher false positive rates for indels. If prioritised variants add up to less than 30 variants, we added passenger mutations, including

even synonymous SNVs. We now added an oncoplot summarizing small variants in the genes *BRAF*, *NRAS*, *TERT*, *CDKN2A*, *RAC1*, *PTEN* and *NF1* (supplementary figure 1). Most tumour-informed panels include variants in known driver genes, however, only few of them are recurrent hotspots. The vast majority of variants included in the panels are patient specific.

Can your method can be simplified to be used in clinical practice? At this point is it possible to be used as it is? How much time is needed from blood collection /tumor collection till having the 30 variants analysed in the blood?

Answer: Yes, the novelty of our method is that we can go from tumour-normal sequencing to the first liquid biopsy results within 4 weeks due to the fast-track design of tumour-informed panels. As soon as the enrichment panel is available, new liquid biopsies can be sequenced quickly reducing processing time to 2 weeks. Liquid biopsies are currently used in the translational setting e.g. for patients of our molecular tumour board. When we start with the tumour-normal sequencing, the first liquid biopsy is archived, and the tumour-informed panel ordered directly after finishing the somatic variant calling for the tumour (approx. 2 weeks for tumour analysis and tumour-informed panel design, 1 week for delivery of panels). By the time the second liquid biopsy arrives, the tumour-informed panel is available and we sequence both liquid biopsies and report the results back to the treating physician. Tumour-informed panels are stored for further time points, leading to reduction in price per time point for extended monitoring. We added a short paragraph to the Discussion (line 259).

Reviewer #3 (Remarks to the Author): expertise in immunotherapy clinical trials

well written and well conducted study

Answer: We thank the reviewer for the valuable suggestions and interesting discussions regarding the current state of the field!

comments:

1. The gold standard in imagin is still CT rather than PET/CT. I recognise there has been more data recently supporting the use of PET/CT in assessing response in melanoma patients and also hypothesising that its use can be useful in better determining response and when a patient may stop therapy within the context of toxicity/other issues with treatment. I think a brief reason why PET/CT was used here as opposed to CT alone would be useful, maybe split ebtween introduction and discussion as appropriate. There is no or little discussion around the issues of response assesmsnet such as pseudoprogression and also stable disease on CT and how this relates in reality to biological response and long term outcome.

Answer: We added more information in regard to the diagnostic value of the gold standard "PET/CT" (line 237).

2. Can the authors comment briefly on how applicable their technique is for use now and how the field may develop;

Answer: Our technology can be used in diagnostics because of the short time for generating tumour-informed panels. We require only four weeks from receiving the primary tumour to generating the results of the first liquid biopsy (pre-treatment) and second liquid biopsy (first LB under treatment, if treatment has started shortly after surgery). These 4 weeks include the time for sequencing and analysing the primary tumour (approx. 2 week), design and delivery of the tumour-informed panel (approx. 1 week) and the sequencing of the cell-free DNA. Follow-up samples can be done even faster, as the tumour-informed sequencing panel is

already available. Despite the technological advances, we see challenges in the implementation of the procedure in clinical patient management (e.g. regular blood draws every visit as well as reimbursement). We expect that the German healthcare system will support routine tumour-liquid liquid biopsies within the next 2-4 years.

standardisation of methodology and how, briefly comment on how comparable techniques are and if there is an evolving common consensus on what constitutes a positive or negative result.

Answer: Standardisation is an important topic and further standardisation is urgently needed. Besides single FDA/EMA approved tests mainly for single mutations, there is a huge number of different technologies available for the analysis of cell-free DNA (including variant and methylation-based methods). Currently, there is no clear consensus standard (e.g. limit of detection, threshold for minimum allele frequencies, minimum coverage of unique molecules per monitored variant, error correction by UMIs, sample quality control) for tumour-informed liquid biopsies. National and international efforts (e.g. German Network for Personalized Medicine - ZPM/DNPM, German Consortium for Translational Oncology - DKTK) in which we participate, as well as the European funded project Can.Heal are underway to define standards for the use of tumour-informed liquid biopsies. We added a comment on this to the discussion (line 262).

The patient graphs they show in figure 2 show that the levels in 027 did go down but then increased again and not to the level of baseline. Is it going to be possible to define categories of responses or is this always going to be an individual response with several readings of ctDNA per patient to make the assessment?

Answer: Based on the observed curves, we can distinguish the following six cases regarding ctDNA-levels: increase, constant, partial decrease, full decrease, partial or full decrease followed by increase, and increase from zero (adjuvant setting). These correspond in many cases to the clinical course. Of special interest are two subtypes which include an initial decrease followed by an increase and an increase from zero. Especially an initial decrease followed by increase could point at an initial response followed by resistance. But it is clear that at this point further studies are needed to establish thresholds and clinical utility (s. question 3 as well).

3. it is intriguing to see the palliative and adjuvant graphs in figure 2 as indicating that ctDNA may be used to tailor treatment such as using single agent ICI then combination if toxicity may be an issue. Such a trial design may be possible along with the genetic markers of IO toxicity or immune cell monitoring eg Bcell work that has been published over the past 2-3 years indicating that these subcells can identify patients prone to IO toxicity.

Answer: This is indeed one of the most important questions in this context and we currently are planning to set up an interventional clinical trial to answer it.

4. I found it hard to follow the use of cfDNA here and ctDNA when they have essentially only looked at ctDNA. is it necessarily to use both terms here?

Answer: as the majority of DNA circulating in plasma does not originate from tumour cells, we use the general term cfDNA (circulating cell-free DNA) for the extracted DNA, which includes DNA from tumour but also other cells. We use ctDNA (circulating cell-free tumour DNA) when we specifically refer to molecules from the tumour. The ctDNA fraction, i.e. the fraction of

ctDNA from all sequenced cfDNA molecules, is an important biomarker for tumour progression and treatment success. We think that it is useful to keep this distinction. However, we re-evaluated the use of the two terms across the manuscript and made the following adjustments: we now mainly refer to “ctDNA” (when speaking about a mutated molecule), “ctDNA fraction/level” when speaking about the ratio of ctDNA / cfDNA, and changed cfDNA to cell-free DNA.

REVIEWER COMMENTS

Reviewer #2 (Remarks to the Author):

The authors addressed my comments; the message is clearer now, and the errors (including % of patients presenting with certain genetic modifications) have been corrected and the approach is now more detailed in order for the reader to have the complete information regarding the choice of the variants. Also the information regarding the time frame and the implementation the clinical settings has been answered.

An additional point:

Despite the complexity and sensitivity of the assay, “In the adjuvant group, relapse could be detected in 53.8% of patients 49 (n=13) with clinical apparent disease recurrence” This means that relapse is not detected in nearly half with clinically apparent disease? The sensitivity needs to be higher if you would consider downscaling on imaging, no?

Reviewer #3 (Remarks to the Author):

I have no further comments. The authors have addressed the previous concerns, in particular the relevance of their technique/methodology and its clinical application or applicability in a routine clinical setting. This strengthens the value of the paper for publication in this journal.

Reviewer #4 (Remarks to the Author):

The manuscript by Schroeder et al. describes a tumour informed approach for melanoma monitoring via plasma ctDNA detection. The work presented is of interest to the oncology community as it provides further support for the value of liquid biopsy for cancer monitoring. There are multiple studies in melanoma showing that the changes in ctDNA early during treatment are indicative of response to treatment and that its increase is associated with relapse, albeit at lower sensitivity. While the authors argue in their rebuttal that the sample size is in line with that of other studies, once we divide in subgroups of palliative and adjuvant therapy, the numbers are small to draw strong conclusions.

1. The abstract seems long and very detailed – please review journal guidelines – ‘an abstract of approximately 150 words’
2. The methods do not clearly communicate how cfDNA libraries were prepared after the selection of the patient-specific mutations. Do you sequence with a large panel but only analyse specific mutations, or do you use a custom probe panel for each patient? In other words, please explain the targeted regions for the cfDNA sequencing.
3. Recommend the raw sequencing data, and bait/target bed files should be made available.
4. umiVar is a novel tool, but the methodology is not well described and lacks any benchmarking against standard tools given it underpins the entire study.
5. In general, the “Sequencing of Cell-free DNA” section could use more detail, eg:

- a. Cycles?
 - b. Demux/Pre-alignment steps?
 - c. Bwa settings
 - d. Does umiVar sort reads by UMI>location or was this with samtools/other?
 - e. umiVar settings/steps
6. TMB is mentioned in several figures but there is no explanation on how this was calculated as there are several different approaches highly dependent on the sequencing panels gene/target coverage.
7. In SuppFigure 1 there are 93 patients while in the text there are only 97 patients in the final analysis. Please add a figure legend describing the data presented. Were the 8 patients with no mutations in the figure included? How can we find out what mutations were tracked in which patient?
8. Notably the frequency of NRAS mutated cases was larger than the reported 20-30%. Can you comment of this.
9. The large majority of cases had TERT mutations. Were they included for ctDNA detection? TERT mutations are usually under-represented in cfDNA and also less efficiently sequenced. It will be worth sharing your experience on this.
10. In line 157 reads: 'Interestingly, a few patients with an extraordinarily high tumour load in PET/CT and high AFs in plasma did not show increase but stable AFs on progression.' Can you please explain where in Figure 2, one can see this exemplified?
11. Figure 4E is not in line with the title of the figure legend, and not well explained in the legend.
12. In the discussion you state that '(ctDNA is) superior to established biomarkers such as S100 or LDH'. What data and statistical analysis supports that conclusion? There is no data showing a correlation of those markers with tumour burden or monitoring comparison head to head. Figure 3 should include these comparisons.

Reviewer #5 (Remarks to the Author):

RESPONSE TO REVIEWERS' COMMENTS

Reviewer #2:

We thank the reviewer for the thorough review of our manuscript.

Despite the complexity and sensitivity of the assay, "In the adjuvant group, relapse could be detected in 53.8% of patients 49 (n=13) with clinical apparent disease recurrence" This means that relapse is not detected in nearly half with clinically apparent disease? The sensitivity needs to be higher if you would consider downscaling on imaging, no?

Answer: Thank you for pointing this out, as this part of the abstract was indeed misleading. The seven patients mentioned here were patients with ctDNA detected before or at the time of relapse. In total, we were able to detect ctDNA in 10 of 13 adjuvant patients with relapse before or within maximally 21 days after relapse (see Fig. 4E). For patient 68 with a positive test at 21 days after relapse, no plasma sample closer to the PET/CT was available. We have clarified this in the abstract and the results section accordingly (starting at line 193). Regarding the reviewer's second question, we agree that the results are not sufficient to reduce imaging tests based on ctDNA results, but the ctDNA results could be used to perform imaging earlier given a positive ctDNA result. Five of the 13 adjuvant patients with relapse were ctDNA positive more than 2 weeks (and up to 133 days) before relapse detection by PET/CT, demonstrating that ctDNA is a promising complementary biomarker for supporting established staging procedures. We have rephrased the paragraph in the results section to make this clearer (lines 184-208).

Reviewer #3:

We thank the reviewer for his suggestions and thorough review of the manuscript.

Reviewer #4:

1. *The abstract seems long and very detailed – please review journal guidelines – ‘an abstract of approximately 150 words’*

Answer: We thank the reviewer for pointing this out. We shortened the abstract to 260 words. Further reduction would, in our opinion, require the removal of crucial information.

2. *The methods do not clearly communicate how cfDNA libraries were prepared after the selection of the patient-specific mutations. Do you sequence with a large panel but only analyse specific mutations, or do you use a custom probe panel for each patient? In other words, please explain the targeted regions for the cfDNA sequencing.*

Answer: We used an individual panel for each patient containing the probes for somatic variants of the patient and additional germline ID SNVs (same for all patients). We added a sentence to the Methods section (lines 335-337).

3. *Recommend the raw sequencing data, and bait/target bed files should be made available.*

Answer: We have added a new supplementary excel file containing all target variants per patient used for monitoring. Due to data protection restrictions and the patient's consent, raw

data unfortunately cannot be made available in public databases. But it will be available to researchers upon reasonable request.

4. *umiVar is a novel tool, but the methodology is not well described and lacks any benchmarking against standard tools given it underpins the entire study.*

Answer: Our umiVar tool is open source and available on github (<https://github.com/imgag/umiVar>, see line 349). Currently, a separate publication is in preparation in which umiVar is benchmarked against other available tools/pipelines using commercial reference samples. The results of the benchmarking study so far show comparable or better performance than similar tools. We believe that this benchmarking effort is out of scope of this manuscript.

5. *In general, the “Sequencing of Cell-free DNA” section could use more detail, eg:*

a. *Cycles?*

b. *Demux/Pre-alignment steps?*

c. *Bwa settings*

d. *Does umiVar sort reads by UMI>location or was this with samtools/other?*

e. *umiVar settings/steps*

Answer: The sequencing was done using different illumina sequencing run protocols with different number of cycles (e.g. 101+10+10+101 or 109+10+10+109). In our experience, slight changes in the numbers of sequencing cycles do not affect the overall results. Demultiplexing was done using the standard illumina pipeline and adapter-trimming with SeqPurge (PMID 27161244) was performed before alignment of reads. BWA-mem2 (<https://github.com/bwa-mem2/bwa-mem2>) was used for mapping with -K 100000000 -Y and indel-realignment was done with abra2 (PMID 30649250). Read de-duplication was done with the barcode correction tool of umiVar with standard settings. We have added this information to the Methods section (lines 348 to 355).

6. *TMB is mentioned in several figures but there is no explanation on how this was calculated as there are several different approaches highly dependent on the sequencing panels gene/target coverage.*

Answer: The tumour sequencing and data analysis pipeline was done as described in our previous study (PMID 35192052). This paper includes the calculation of the TMB. We have adjusted the methods section accordingly (line 310).

7. *In SuppFigure 1 there are 93 patients while in the text there are only 97 patients in the final analysis. Please add a figure legend describing the data presented. Were the 8 patients with no mutations in the figure included? How can we find out what mutations were tracked in which patient?*

Answer: We thank the reviewer for this comment. We believe the reviewer refers to the final cohort of 87 patients (not 97), compared to the 93 patients included in the oncoplot. In total, 6 patients were removed as the quality control of their liquid biopsies failed and no other samples were available for these patients. Nonetheless, results from tumour-normal

sequencing of all 93 patients were included in the oncoplot (Supplementary Fig. 2). Moreover, the oncoplot only shows hotspot and non-hotspot variants in a subset of known driver genes. During the design process of the individual tumour-informed panels for each patient, both driver and passenger variants were selected, and our approach works even if only passenger mutations are identified for a patient. Therefore, the absence of driver mutations in the oncoplot does not indicate that the panel design was not possible.

However, three of the patients without driver mutations were indeed excluded because they had only few variants lying in regions which showed bad enrichment or sequencing quality (e.g. in homopolymers, indels, very low or very high GC content regions). We note that this issue can easily be avoided by sequencing tumours with WES instead of a large gene panel, as the extended target regio of WES always provides >30 somatic variants (see lines 278-282).

8. Notably the frequency of NRAS mutated cases was larger than the reported 20-30%. Can you comment of this.

Answer: We assume that the reviewer's question is related to the differences in the number of *NRAS* variants stated in line 121 compared to the oncoplot in Supplementary Fig. 2. We only refer to hotspot mutations in the results section, whereas the oncoplot also includes other mutations in the same gene. The reason for referring to hotspots is that typically hotspot mutations are used in commercial liquid biopsy panels (e.g. droplet-based digital PCR). Our approach can use any patient-specific variant and is therefore not restricted to patients with hotspot mutations.

9. The large majority of cases had TERT mutations. Were they included for ctDNA detection? TERT mutations are usually under-represented in cfDNA and also less efficiently sequenced. It will be worth sharing your experience on this.

Answer: We thank the reviewer for this relevant question. *TERT* promoter variants were included in the design of tumour-informed panels of 26 patients. We observed no difference in the performance of promoter variants compared to coding, intronic or regulatory variants of other genes. A supplementary figure with two boxplots was added showing the sequencing depth of *TERT* promoter variants compared to the other monitored variants after deduplication (Supplementary Fig. 1). A reference to the promoter variants in *TERT* and the sequencing performance was added to the manuscript (lines 122-124).

10. In line 157 reads: 'Interestingly, a few patients with an extraordinarily high tumour load in PET/CT and high AFs in plasma did not show increase but stable AFs on progression.' Can you please explain where in Figure 2, one can see this exemplified?

Answer: Figure 2 does not include patients with the combination of high but stable AFs and progressive disease. We have now added an additional supplementary figure (Supplementary Fig. 4) with an example of a patient with high AFs at the beginning of treatment, which remained stable during treatment while the patient was diagnosed with progressive disease. In our opinion, this corresponds to a saturation of the cell-free DNA with ctDNA. The new supplementary figure is referenced in the manuscript in line 143.

11. *Figure 4E is not in line with the title of the figure legend, and not well explained in the legend.*

Answer: We adjusted the figure title and extended the legend of subpanel E.

12. *In the discussion you state that '(ctDNA is) superior to established biomarkers such as S100 or LDH'. What data and statistical analysis supports that conclusion? There is no data showing a correlation of those markers with tumour burden or monitoring comparison head to head. Figure 3 should include these comparisons.*

Answer: Figure 3 does not include this information. We have now added a Venn diagram in the supplement comparing liquid biopsies to S100 and LDH at T0 for the advanced melanoma group (Supplementary Fig. 6). The majority of biomarker-positive patients was found to be ctDNA-positive (11 positive patients only in ctDNA), with only one patient positive for S100/LDH and ctDNA-negative. The new supplementary figure is referenced in the manuscript in line 160. We changed the results section accordingly and removed the comparison of allele frequencies between the different biomarkers in our previous version (lines 159-161).

Reviewer #5:

We thank the reviewer for his comments and suggestions.

REVIEWERS' COMMENTS

Reviewer #2 (Remarks to the Author):

The authors addressed the comments/corrections.
No further observations.

Reviewer #4 (Remarks to the Author):

The authors have addressed the comments raised and made the appropriate changes to the manuscript, which is now more transparent and easier to follow.

Reviewer #5 (Remarks to the Author):
